# Visible-Light-Active N-Doped TiO_2_ Photocatalysts: Synthesis from TiOSO_4_, Characterization, and Enhancement of Stability Via Surface Modification

**DOI:** 10.3390/nano12234146

**Published:** 2022-11-23

**Authors:** Nikita Kovalevskiy, Dmitry Svintsitskiy, Svetlana Cherepanova, Stanislav Yakushkin, Oleg Martyanov, Svetlana Selishcheva, Evgeny Gribov, Denis Kozlov, Dmitry Selishchev

**Affiliations:** 1Department of Unconventional Catalytic Processes, Boreskov Institute of Catalysis, Novosibirsk 630090, Russia; 2Department of Heterogeneous Catalysis, Boreskov Institute of Catalysis, Novosibirsk 630090, Russia; 3Department of Physicochemical Methods of Research, Boreskov Institute of Catalysis, Novosibirsk 630090, Russia; 4Engineering Center, Boreskov Institute of Catalysis, Novosibirsk 630090, Russia

**Keywords:** photocatalytic oxidation, visible light, N-doped TiO_2_, copper (Cu), nanocomposite, action spectrum, EPR, stability

## Abstract

This paper describes the chemical engineering aspects for the preparation of highly active and stable nanocomposite photocatalysts based on N-doped TiO_2_. The synthesis is performed using titanium oxysulfate as a low-cost inorganic precursor and ammonia as a precipitating agent, as well as a source of nitrogen. Mixing the reagents under a control of pH leads to an amorphous titanium oxide hydrate, which can be further successfully converted to nanocrystalline anatase TiO_2_ through calcination in air at an increased temperature. The as-prepared N-doped TiO_2_ provides the complete oxidation of volatile organic compounds both under UV and visible light, and the action spectrum of N-doped TiO_2_ correlates to its absorption spectrum. The key role of paramagnetic nitrogen species in the absorption of visible light and in the visible-light-activity of N-doped TiO_2_ is shown using the EPR technique. Surface modification of N-doped TiO_2_ with copper species prevents its intense deactivation under highly powerful radiation and results in a nanocomposite photocatalyst with enhanced activity and stability. The photocatalysts prepared under different conditions are discussed regarding the effects of their characteristics on photocatalytic activity under UV and visible light.

## 1. Introduction

Currently, heterogeneous photocatalysis using semiconducting nanomaterials is one of the key research areas worldwide, because it is regarded as an efficient method for utilizing solar radiation to carry out useful chemical transformations and store energy [1,2]. Since 1972, when Fujishima and Honda published their study on electrochemical water splitting using a semiconductor electrode [3], titanium dioxide has been a focus of attention as an active and stable material for various photocatalytic reactions [4,5,6]. The great importance of TiO_2_ as a photocatalyst is also due to the large reserves of titanium minerals in the Earth’s interior and the relatively low cost of its manufacturing.

In an oxygen-containing medium, photocatalytically-active TiO_2_ can decompose hazardous molecular pollutants under UV light [7,8]. This behavior enables the application of TiO_2_-mediated photocatalytic oxidation as an efficient and environmentally friendly method for the purification of air and water [9,10,11]. At the same time, a substantial hindrance for the application of TiO_2_ photocatalysts is that titanium dioxide has a wide band gap (e.g., 3.2 eV for anatase) and can only absorb UV light. The control of mercury, especially in light sources, the high cost of UV light-emitting diodes (UV LEDs) as an alternative to mercury lamps, and the low content (<5%) of UV light in solar radiation requires the development of efficient photocatalysts to carry out photocatalytic reactions under wide-range radiation, including visible light.

Narrow-band semiconductors (e.g., graphitic carbon nitride, silver phosphate, bismuth vanadate, ternary sulfides) exhibit photocatalytic activity under visible light but quantum efficiencies in oxidation reactions over these single-phase materials are commonly not high enough due to surface properties, nonoptimal positions of energy bands, and a high recombination rate of photogenerated charge carriers [12]. For these reasons, the modification of TiO_2_ to extend its action spectrum to the visible region is still the main way to develop an efficient photocatalyst for the degradation of pollutants [13,14].

The absorption of visible light and photoexcitation of electrons in TiO_2_-based photocatalysts can be achieved by the sensitization of TiO_2_ with organic dyes or plasmonic metals; the combination of TiO_2_ with narrow-band semiconductors can be used to form heterojunctions, as well as TiO_2_ doping with metals and nonmetals for the creation of additional energy levels in its band gap [15,16,17,18,19]. Creating bulk/surface defects (e.g., reduced states, oxygen vacancies) in semiconducting materials, including TiO_2_, and tuning their concentrations are also strategies that allow an increase in the absorption of visible light and an enhancement of the photocatalytic activity [20].

Concerning the approaches mentioned above, TiO_2_ doped with nitrogen can exhibit great activity in the photodegradation of pollutants both under UV and visible light. In 1986, Sato et al. [21] first described an N-doped TiO_2_ photocatalyst prepared via the precipitation of titanium hydroxide using ammonia solution, which was able to oxidize ethane and CO under visible radiation (λ_max_ = 434 nm). In 2001, Asahi et al. [22] reported the photocatalytic activity of N-doped TiO_2_ in the degradation of methylene blue and acetaldehyde vapor under similar radiation (436 nm). The authors suggested that nitrogen substitutes oxygen in lattice sites of TiO_2_, and N 2p orbitals are mixed with O 2p orbitals from the valence band (VB) of TiO_2_ that narrows its band gap and results in the ability for absorption of visible light. On the other hand, Di Valentin et al. [23,24,25] have shown in a series of studies using computer simulation and EPR techniques that no narrowing of the band gap occurs during TiO_2_ doping with nitrogen, whereas discrete energy levels appear above the VB of TiO_2_. The authors reported that nitrogen can substitute O atoms in the lattice sites (i.e., substitutional position) with the formation of N 2p discrete levels located 0.14 eV above the VB or can occupy the interstitial sites (i.e., interstitial position) with the formation of π * NO discrete levels located 0.73 eV above the VB.

The position occupied by nitrogen depends on the preparation method of N-doped TiO_2_. Various chemical compounds were used as a source of nitrogen during the precipitation of TiO_2_: ammonium hydroxide [26,27], ammonium chloride [24,28,29], ammonium nitrate [25], nitric acid [25,30], hydrazine [25,31], urea [32,33], diethanolamine [32], and triethylamine [31,32]. A shift of light absorption edge to the visible region compared to pristine TiO_2_ was observed for all photocatalysts synthesized using the mentioned precursors. The photocatalytic activity of N-doped TiO_2_ has been commonly investigated in the degradation of organic contaminants in air or water using batch reactors. Data on the visible-light-driven degradation of formic acid [34], acetaldehyde [22], phenol [27], 2-chlorophenol [32], 2,4-dichlorophenol [35], stearic acid [36], methylene blue [28,37,38,39,40], rhodamine B [35], eriochrome black-T [40], phenanthrene [41], dibenzothiophene [42], ethylparaben [43], lindane [44], atrazine [45], mecoprop and clopyralid herbicides [46] have been reported. The ability to decompose organic contaminants under irradiation was demonstrated, but many studies did not provide the data, which can be used for comparison of the results between each other (e.g., quantum efficiencies or data for benchmarks). Additionally, it is difficult to evaluate the steady-state photocatalytic activity in a batch reactor because the concentration of oxidizing contaminant is decreased monotonically during the experiment and, consequently, affects the reaction rate. Experiments in a continuous-flow reactor are preferable to receive valid information on the efficiency of light utilization and the stability of photocatalysts.

Despite the superiority of titanium alkoxide precursors in scientific papers, one of the two main methods for the production of TiO_2_ in industry is sulfate technology using ilmenite mineral (FeTiO_3_), which results in the formation of titanium (IV) oxysulfate (TiOSO_4_) followed by its hydrolysis to titanium (IV) oxide hydrate (TiO_2_ × nH_2_O). Little information on the application of inorganic titanium precursors for the preparation of N-doped TiO_2_ photocatalysts can be found in the literature. It was our motivation to study the preparation of N-doped TiO_2_ photocatalysts using titanium oxysulfate as a titanium precursor and ammonia as a nitrogen source. In contrast to titanium alkoxide precursors, titanium oxysulfate can be easily dissolved and hydrolyzed in aqueous solutions without organic solvents.

In this study, the preparation of highly active N-doped TiO_2_ photocatalyst was performed via precipitation from an aqueous solution of titanium oxysulfate using ammonium hydroxide followed by thermal treatment in air. Despite doubts about TiO_2_ doping with nitrogen during wet synthetic processes [47], we show experimental data that clearly confirm this mechanism of modification in the case of photocatalysts prepared from TiOSO_4_ using ammonia. The effect of preparation conditions on the characteristics and photocatalytic activity of N-doped TiO_2_ is discussed. A comparison with commercially available photocatalysts is also made. An approach via the surface modification of N-doped TiO_2_ with copper species is proposed for the preparation of nanocomposite photocatalysts with an enhanced stability under long-term exposure to power radiation.

## 2. Materials and Methods

### 2.1. Chemicals

Ammonium hydroxide solution (NH_4_OH, 25%) and sodium hydroxide (NaOH, reagent grade) from AO Reachem Inc. (Moscow, Russia), titanium(IV) oxysulfate dihydrate (TiOSO_4_ × 2H_2_O, reagent grade) and copper(II) acetate monohydrate (Cu(CH_3_COO)_2_ × H_2_O, 99%) from AO Vekton Inc. (Moscow, Russia), and iron(III) nitrate nonahydrate (Fe(NO_3_)_3_ × 9H_2_O, 98%) from Sigma–Aldrich (St. Louis, MO, USA) were used for the preparation of photocatalysts. Reagent grade acetone (CH_3_COCH_3_) from Mosreaktiv LLC (Moscow, Russia) was used as a test organic compound for oxidation during the photocatalytic experiments. Two commercially available powdered TiO_2_ photocatalysts, namely, TiO_2_ P25 from Evonik Ind. AG (Essen, Germany) and TiO_2_ KRONOClean 7000 from KRONOS Worldwide Inc. (Dallas, TX, USA), were employed as the references for benchmarking. These photocatalysts are referred to in this paper as P25 and KC, respectively. All chemicals and materials were used as received from suppliers without further purification.

### 2.2. Preparation of Photocatalysts

The samples of N-doped TiO_2_ (TiO_2_-N) were prepared via the precipitation method using TiOSO_4_ as a titanium precursor and ammonium hydroxide as a precipitating agent, as well as a source of nitrogen. Before the synthesis, titanium oxysulfate dihydrate (83 g) was suspended in deionized water (600 mL) and stirred for several days. A small amount of undissolved precursor was removed from the reaction solution using a 0.20 µm PTFE membrane filter. A commercial solution of ammonium hydroxide was diluted 10 times with deionized water. In a typical synthesis, prepared solutions of TiOSO_4_ (ca. 200 mL) and ammonia were simultaneously added dropwise to deionized water (1000 mL) under vigorous stirring. A pH electrode was immersed into the reaction solution to adjust the pH during the precipitation to 7 by tuning the flows of solutions. The white precipitate formed after mixing the reagents was aged in mother liquor for several days. Then, the precipitate was separated by centrifugation at 8000 rpm for 10 min and washed with deionized water. The centrifugation and washing procedures were repeated up to 15 times until the conductivity of washouts became similar to that of pure water. Finally, the precipitate was dried and further calcined in air for 3 h at a certain temperature in the range of 250–600 °C. In addition to TiO_2_-N, blank TiO_2_ without added nitrogen was prepared by similar technique but using sodium hydroxide as a precipitating agent. The precipitation was performed at pH = 7 followed by aging and calcination at 350 °C.

One of the prepared TiO_2_-N samples (350 °C) was further modified with iron or copper species by impregnation with an aqueous solution of iron nitrate (0.15 M) or copper acetate (0.15 M). Typically, an aliquot of iron or copper precursor was added to the suspension of TiO_2_-N (1 g) in deionized water (10 mL). After vigorous stirring for 1–2 h, the sample was separated by centrifugation at 8000 rpm for 10 min and washed 3 times with deionized water. Finally, it was dried in air at 70 °C. The iron and copper contents were 0.1 and 1% of the mass of TiO_2_, respectively.

### 2.3. Characterization Techniques

The phase composition of the photocatalysts was determined by powder X-ray diffraction (XRD) using a D8 Advance diffractometer (Bruker, Billerica, MA, USA) equipped with a CuK_α_ radiation source and a LynxEye position sensitive detector. The data were collected in the 2*θ* range of 5–75° with a step size of 0.05° and a collection time of 3 s. The average size of TiO_2_ crystallites and strain values were estimated from the peak profile parameters (i.e., weight and width of Gauss and Lorentz peak components) determined by double-Voigt analysis in the Le Bail fitting method realized in TOPAS software (Bruker, 4.2 version). Lattice constants were also refined with the mentioned TOPAS software. The morphology of the photocatalysts was studied by SEM using an ultrahigh resolution Field-Emission SEM (FE-SEM) Regulus 8230 (Hitachi, Tokyo, Japan) at an accelerating voltage of 30 kV. The surface composition was investigated by X-ray photoelectron spectroscopy (XPS) using an ES300 spectrometer (Kratos Analytical, Manchester, UK) equipped with a MgKα (h*ν* = 1253.6 eV, 170 W) radiation source. The photoelectron lines of Au 4f_7/2_ (84.0 eV) and Cu 2p_3/2_ (932.7 eV) from metallic gold and copper foils were used to calibrate the scale of binding energy (BE). The C 1s line (284.8 eV) from carbon deposits was used as an internal reference. Pretreatment of samples was performed for 1 h before the acquisition of survey and precision spectra. The surface composition was also investigated using diffuse reflectance infrared Fourier-transform spectroscopy (DRIFTS). The spectra were collected at room temperature under air exposure using a Vector 22 FTIR spectrometer (Bruker) equipped with an EasiDiff^TM^ accessory from PIKE Technologies Inc. (Madison, WI, USA). No pretreatment of the samples was performed before the analysis.

The textural properties were investigated by adsorption of nitrogen at 77 K using an ASAP 2400 instrument (Micromeritics, Norcross, GA, USA). Before the experiments, the samples were thermally treated in vacuum at 150 °C for 16–24 h. The surface area and average pore diameter were calculated by BET analysis of the measured isotherms, and the pore volume was determined as the total pore volume at P/P_0_~1. The total content of nitrogen and sulfur in the prepared photocatalysts was measured by CHNS analysis using a Vario EL Cube elemental analyzer from Elementar Analysensysteme GmbH (Langenselbold, Germany). Five independent measurements were made for each sample, and an averaged value of N and S content was calculated.

UV–vis diffuse reflectance spectra (DRS) were recorded at room temperature in the range of 250–850 nm with 1 nm of resolution using a Cary 300 UV–vis spectrophotometer from Agilent (Santa Clara, CA, USA) equipped with a DRA-30I diffuse reflectance accessory and special prepacked polytetrafluoroethylene (PTFE) reflectance standard. The optical band gap was estimated using the Tauc method based on the assumption of indirect allowed transitions by plotting ${\left(F\left(R\right)h\vartheta \right)}^{\frac{1}{2}}$ versus hϑ and a linear fitting [48]. To analyze the effect of calcination temperature on light absorption in a series of TiO_2_-N samples, the UV–vis spectrum of the TiO_2_ sample was subtracted from the spectra of synthesized TiO_2_-N samples. The received difference spectra were integrated in the region of 380–550 nm, and the integral values were plotted versus the values of calcination temperature used during the synthesis.

Electron paramagnetic resonance (EPR) spectra were recorded using an X-band EleXsys 500 spectrometer (Bruker) equipped with a digital temperature control system ER 4131 VT. The samples for study were placed in a quartz tube at the center of a rectangular TE_102_ resonator with a magnetic component of the UHF field perpendicular to the direction of the external magnetic field. The blue light (450 nm) was supplied directly to the microwave cavity in the process of spectra registration. The spectra simulation was carried out using the EasySpin software (an open-source software package, 5.2 version) [49].

The samples for photoelectrochemical measurements were prepared via the deposition of TiO_2_-N on a fluorinated tin oxide (FTO) glass from an aqueous suspension containing Nafion^TM^ followed by drying at 50 °C overnight and at 150 °C for 30 min. After that, the sample with a spot area of 2 cm^2^ was placed in a photoelectrochemical cell and irradiated with UV light (λ_max_ = 365 nm, 10 mW cm^−2^) or blue light (λ_max_ = 450 nm, 160 mW cm^−2^) provided by light-emitting diodes (LEDs) through the back side of FTO glass and served as a working electrode. A mercury-sulfate electrode and platinum foil served as reference and counter electrodes, respectively. Before the photoelectrochemical tests, the potential of the mercury-sulfate electrode was measured relative to the reversible hydrogen electrode (RHE). The experiments were carried out at room temperature in 1 M Na_2_SO_4_ or 1 M Na_2_SO_4_ + 0.82 M CH_3_OH solution as an electrolyte using an Autolab PGStat 300N potentiostat (Metrohm AG, Herisau, Switzerland). The photocurrent was determined under corresponding radiation by chronoamperometry at a potential of 0.585 V RHE in three light-off cycles.

### 2.4. Photocatalytic Experiments

The synthesized and reference photocatalysts were tested in the oxidation of acetone vapor in a continuous-flow setup designed for the evaluation of steady-state oxidation rates and stability of photocatalytic performance. The details on the experimental setup and evaluation of photocatalytic activity can be found in our previously published paper [50]. Highly powerful UV (λ_max_ = 371 nm) and blue (λ_max_ = 450 nm) LEDs were used for the irradiation of photocatalysts and evaluation of their activity in the UV and visible regions, respectively. The emission spectra of the LEDs used in this work are shown in Appendix A. According to the measurements using an ILT 950 spectroradiometer (International Light Technologies Inc., Peabody, MA, USA), the specific irradiance was 9.7 and 164 mW cm^−2^ for UV and blue light, respectively. The experiments were performed under optimized operational conditions: the area density of the photocatalyst was 19 mg cm^−2^, the irradiation area was 9.1 cm^2^, the inlet concentration of acetone was ca. 32 µmol L^−1^, the reactor temperature was 40 ± 0.1 °C, the relative humidity was 19 ± 1%, and the volume flow rate was 67 ± 1 cm^3^ min^−1^.

The setup had a special valve system for the alternate analysis of the inlet and outlet reaction mixtures using an FTIR spectrometer FT-801 (Simex LLC, Novosibirsk, Russia) equipped with an IR long-path gas cell (Infrared Analysis Inc., Anaheim, CA, USA). The quantitative analysis of acetone and CO_2_ was performed using the Beer–Lambert law by the integration of collected IR spectra in corresponding regions [50]. Appendix A shows a typical CO_2_ concentration profile during the photocatalytic experiment. The photocatalytic activity of the samples was evaluated as the rate of CO_2_ evolution, resulting from the oxidation of acetone. The CO_2_ rate was estimated in continuous-flow experiments as the difference in the outlet and inlet CO_2_ concentrations multiplied by the volume flow rate. After the achievement of a steady-state regime, the CO_2_ rate was averaged for 30 min. Based on the statistics of many experiments, the total error in measuring the CO_2_ formation rate using the setup did not exceed 10%. In the tests on stability, the photocatalytic activity of the samples was evaluated for 30 h.

Additionally, the action spectrum of TiO_2_-N obtained as the dependence of photonic efficiency on the wavelength of excited radiation was also investigated using the mentioned setup. Eighteen LEDs with different basic wavelengths from 365 to 730 nm were used for irradiation of the photocatalyst during the measurement of the action spectrum. The photon flux for each LED was adjusted to the value of ca. 7 × 10^17^ s^−1^ for correct evaluation of efficiency. The photonic efficiency of the photocatalytic reaction can be expressed according to IUPAC recommendations as follows [51]:(1)$$\xi =\frac{W_{r}}{q_{p}^{0}}=\frac{16}{3}\times \frac{W_{CO_{2}}\times 10^{16}}{7\times 10^{17}}\times 100\%$$
where ξ is the photonic efficiency (%), $W_{r}$ is the reaction rate (s^−1^), $q_{p}^{0}$ is the incident photon flux (s^−1^), $W_{CO_{2}}$ is the steady-state rate of CO_2_ formation measured in the setup (µmol min^−1^), 7 × 10^17^ is the photon flux (s^−1^), and $\frac{16}{3}$ is the conversional factor (16 charge pairs and, consequently, at least 16 photons are required to oxidize 1 molecule of acetone; 3 molecules of CO_2_ are formed from 1 molecule of acetone during 1 act of reaction). The details on the measurement of action spectra can be found in recently published papers [52,53].

## 3. Results and Discussion

In this study, we investigated N-doped TiO_2_ photocatalysts synthesized using titanyl sulfate as the titanium precursor and ammonium hydroxide as a precipitating agent, as well as a source of nitrogen. The results on the state of nitrogen in doped TiO_2_ and the effect of preparation conditions on the characteristics and photocatalytic activity of materials are discussed first. Then, the effect of Fe and Cu modifiers on the stability of composite photocatalysts is shown.

### 3.1. Effect of Preparation Conditions on the Characteristics and Photocatalytic Activity of N-Doped TiO_2_

pH-controlled hydrolysis of TiOSO_4_ using an aqueous solution of ammonia leads to the precipitation of titanium oxide hydrate. According to the results of XRD analysis, this precipitate was completely amorphous and had no photocatalytic activity under visible light. The corresponding XRD patterns of the samples calcined at different temperatures are shown in Figure 1a.

The peaks, which can be attributed to the crystal phase of anatase, were detected in the patterns only at 250 °C, but the size of crystallites in the corresponding sample was extremely low for quantitative evaluation. The main peaks at 2θ of 25.3°, 37.8°, 48.1°, 53.9°, 55.1°, and 62.8° attributed to the (101), (004), (200), (105), (211), and (204) planes of anatase TiO_2_ clearly indicate the formation of the anatase phase at an increased temperature (350 °C and higher). The average crystallite size in the sample calcined at 350 °C was estimated to be 16 nm. The SEM micrographs in Figure 1b and c illustrate this statement and show that large particles of synthesized TiO_2_-N consist of agglomerated nanocrystallites of an oval form.

It should be noted that the consideration of only the size effect during the analysis of XRD patterns did not give accurate fitting of the experimental patterns of synthesized samples (see Appendix A). Therefore, the estimation of crystallite size was performed considering the effect of strains according to the Le Bail fitting method. Table 1 shows the estimated strain values that are high enough (ε > 0.001) for all synthesized samples. This fact indicates the presence of microdeformations in the structure of TiO_2_ nanocrystals that can be associated with impurities. According to CHNS analysis, the synthesized TiO_2_-N contains nitrogen and sulfur as the main impurities. The content of both elements depended on the preparation conditions (Table 1). The lattice constants (a and c) and, consequently, the volume of the unit cell for this TiO_2_-N sample substantially differ from the corresponding parameters for the TiO_2_ sample synthesized without the addition of ammonia (Appendix A), which also confirms a key effect of nitrogen impurities.

An increase in the calcination temperature led to an increase in the size of crystallites and the extent of crystallinity because of a decrease in the value of strains (Table 1). A monotonic decrease in the stain values confirms reducing microdeformations in the structure of TiO_2_ nanocrystals during calcination at higher temperatures. It should be noted that even calcination at the highest temperature (600 °C) did not lead to the transformation of anatase to the rutile phase because no peaks attributed to rutile were detected in the corresponding XRD pattern (Figure 1a). This is a common case for photocatalysts prepared from the TiOSO_4_ precursor due to the stabilization of the TiO_2_ surface by strongly bonded sulfates [54]. Typically, much higher temperatures are required for anatase–rutile transformation in this system.

The calcination temperature strongly affected the textural properties of the materials. Table 1 shows that the specific surface area and pore volume of the samples decrease as the calcination temperature increases. In contrast, the average diameter of pores increases as the temperature increases, which agrees with the growth of crystallite sizes. Sintering TiO_2_ nanoparticles occurs during calcination, which results in the formation of large agglomerates with a lower specific surface area. Table 1 also shows that an increase in the calcination temperature leads to a monotonic decrease in the content of nitrogen estimated using CHNS analysis. This confirms the degradation of nitrogen species due to their oxidation at increased temperatures followed by their removal from the material. At the same time, the sulfur content has a similar value over the whole temperature range (Table 1). Sulfur species are commonly more stable on the surface of TiO_2_ than nitrogen species due to a higher affinity [55], and much higher temperatures are required for their complete removal.

The synthesized samples had a yellow color, which confirms their ability to absorb light in the visible region of the spectrum. The UV–vis spectrum of TiO_2_-N shows a shoulder of additional absorption in the region of 400–540 nm compared to the fundamental absorption of TiO_2_ (Figure 2a). This behavior can be attributed to nitrogen impurities in TiO_2_ rather than sulfur impurities because the TiO_2_ sample, prepared similarly from the TiOSO_4_ precursor using NaOH as a precipitating agent, has no absorption in the corresponding region. As expected, the TiO_2_ sample is white in color, in contrast to TiO_2_-N (see photographs in Figure 2a).

UV–vis spectra of commercially available TiO_2_ P25 and TiO_2_ KC photocatalysts are also shown in Figure 2a for comparison. TiO_2_ P25 consists of both anatase and rutile crystal phases, and it can absorb radiation with wavelengths up to 410 nm. At the same time, TiO_2_ KC can absorb light in the visible region up to 750 nm, but its absorption spectrum differs from that of TiO_2_-N due to another type of modification [56].

The optical band gaps of synthesized and commercial photocatalysts, estimated using the Tauc method, were in the range of 3.2–3.3 eV (Figure 2b, Table 1), which can be attributed to the anatase phase of TiO_2_. A value of 3.05 eV can also be estimated from the Tauc plot for the TiO_2_ P25 photocatalyst. This value can be attributed to the rutile phase in the P25 composition. The inset in Figure 2b shows that in the case of TiO_2_-N we can also estimate the minimum energy required for the photoexcitation of electrons to the CB of TiO_2_. A linear approximation of the additional absorption shoulder gives a value of 2.3 eV that corresponds to 540 nm. As mentioned above, the absorption of light in the visible region is due to nitrogen impurities, which create additional energy levels in the band gap of TiO_2_ and, consequently, decrease the minimum energy required for the excitation of electrons.

Figure 3a shows that the calcination temperature strongly affects the UV–vis spectra of synthesized TiO_2_-N samples. This effect was considered by the integration of difference spectrum between TiO_2_-N and TiO_2_. The amount of light absorbed by the TiO_2_-N samples in the region of 380–550 nm had a dome-shaped dependence on the calcination temperature with a maximum at 400 °C (Figure 3b). This indicates that at low calcination temperatures, the samples absorb a small amount of light, but very high temperatures (550–600 °C) also result in weak light absorption. The samples are to be calcined at an optimal temperature (350–450 °C) to absorb the highest amount of light.

At the same time, Table 1 shows that the band gap and the minimum energy required for photoexcitation of TiO_2_-N samples have similar values over the whole range of temperatures. This means that the temperature of sample calcination in air has a low impact on the type of doping, but it strongly affects the amount of nitrogen impurities that are responsible for the absorption of visible light.

According to a previously published study [25], the mentioned absorption can result from the nitrogen species, which occupy the interstitial positions in the TiO_2_ lattice. To verify this hypothesis, X-ray photoelectron spectroscopy was used for the analysis of the nitrogen state in TiO_2_-N.

Figure 4a shows a typical photoelectron N 1s spectral region for the synthesized TiO_2_-N photocatalyst. This N 1s spectrum can be fitted into two components with binding energies (BEs) of 397.7 eV (N1 state) and 399.5 eV (N2 state). The N2 state gives a contribution higher than 90%. Commonly, the BE (N 1s) in the range of 396–398 eV can be attributed to negatively charged forms of nitrogen, which substitute lattice oxygen in the TiO_2_ structure (i.e., located in the substitutional positions) [57,58,59,60]. A variation in BE for this type of nitrogen can be attributed to the localization of nitrogen (on the surface/in the bulk) [61] and its additional bonding with oxygen [62]. The N2 state with a BE (N 1s) of 399.5 eV might be attributed to weakly charged nitrogen species, which result from the interactions of nitrogen with C, H, O atoms [36,63,64], or to partially oxidized nitrogen located in the interstitial positions of the TiO_2_ lattice [65,66,67]. According to a computer simulation study [24], the difference between the BE values for nitrogen in substitutional and interstitial positions is 1.6 eV, which is close to the difference between the maximums of N 1s peaks attributed to the N1 and N2 components in our case (ΔE = 1.8 eV). This also supports the interpretation of the detected N1 and N2 states. Notably, the chemical state of nitrogen in nitrites (${\mathrm{NO}}_{2}^{-}$) and nitrates (${\mathrm{NO}}_{3}^{-}$) is characterized by a BE (N 1s) value higher than 403 eV. In our case, no bands in this region were detected in the N 1s spectrum of TiO_2_-N, which indicates the absence of nitrites and nitrates on the surface of the studied sample after vacuum treatment. Therefore, XPS analysis of the synthesized TiO_2_-N photocatalyst shows that nitrogen incorporated into interstitial positions of TiO_2_ gives the major contribution. It is important to note that a small peak at 399.6 eV was also observed in the N 1s spectral region of the TiO_2_ sample synthesized without the addition of ammonia (Figure 4c), but this low-intensity signal can appear due to nitrogen species in a support used for fixation of the sample in the spectrometer chamber.

Concerning the chemical state of titanium, the position of the Ti 2p_3/2_ peak in the photoelectron Ti 2p spectral region (Figure 4b) is characterized by a BE value of 458.3 eV, which corresponds well to the Ti^4+^ state in TiO_2_. The Ti 2p spectrum has a doublet structure due to spin-orbit splitting (ΔE = 5.7 eV), and corresponding shake-up satellites with ΔE ~ 13 eV are also observed. An additional component with a BE value of 456.5 eV appeared during the curve fitting of the Ti 2p spectrum and was attributed to traces of reduced Ti species (first, Ti^3+^) stabilized near structural defects [68,69]. The surface concentration of these Ti^3+^ species was no more than 4%. The TiO_2_ sample synthesized using NaOH as the precipitating agent had similar peaks in the Ti 2p spectral region and the same concentration of reduced Ti species (Figure 4d).

In addition to XPS analysis, the prepared photocatalysts were studied using the EPR technique. An EPR signal with a g-factor of 1.99 and dH = 3 mT was observed for the synthesized TiO_2_ sample, which had no nitrogen impurities. This signal was insensitive to irradiation of the sample with light. Kumar et al. [70] attributed the signal with similar parameters to localized electron centers (such as Ti^3+^) in the TiO_2_ lattice. The synthesized TiO_2_-N sample had the same signal under dark conditions. At the same time, exposure of this sample to blue light in situ during the registration of spectra led to a change in the form of the signal because three narrow lines appeared. This indicates the presence of additional types of paramagnetic centers in TiO_2_-N. Light allows an increase in the concentration of these centers up to saturation.

Figure 5a shows the EPR spectra of TiO_2_-N in the dark and under blue light. A difference spectrum, obtained via the subtraction of the dark spectrum from the spectrum under light, represents three lines with a hyperfine splitting of 3.22 mT that corresponds well to the nucleus of nitrogen. The parameters of the spectrum simulation are shown in Table 2. They agree with the literature data and can be attributed to N-based light-induced paramagnetic species.

It is important to note that the intensity of signals for these paramagnetic species decreased monotonically when the sample was kept in the dark. Figure 5b shows that the rate of decay increases as the temperature of the sample increases. The decay curves were analyzed using the stretched exponential function:(2)$$\xi I\left(t\right)=I_{0}+A\cdot exp\left(-{\left(t/\tau \right)}^{\beta }\right)$$
where τ is the average decay time and *β* is the decay exponent, derived from the fact that the signal decay of photoinduced defects can be characterized by the distribution of time constants rather than a single constant, which is common for amorphous materials [71]. The value of the decay exponent *β* used in the estimation was 0.5. In this assumption, the lifetime has an activated behavior and can be described as τ(T)~exp(ΔE/T) [72]. A linearization in the Arrhenius coordinates gives an activation energy of 0.42 eV.

Summarizing the data of UV–vis, CHNS, XPS, and EPR techniques, the N-doped TiO_2_ synthesized from the TiOSO_4_ precursor using ammonia can absorb visible light due to the oxidized nitrogen species, which are localized in the interstitial positions in the TiO_2_ lattice.

Figure 6a shows that under UV and blue light irradiation, the synthesized TiO_2_-N can generate a current with a density up to 15 μA cm^−2^. These data indicate that photogeneration and interface transfer of charge carriers occur in TiO_2_-N under exposure to radiation of both spectral regions corresponding to intrinsic and extrinsic absorptions of light. In addition, the visible-light induced photocurrent of TiO_2_-N doubled when methanol was added to the electrolyte. An increased photocurrent in the presence of methanol can be attributed to an improved pathway of photogenerated holes due to their transfer from TiO_2_-N to the organic donor of electrons (i.e., methanol), which enhances the separation of charge carriers. The results of photoelectrochemical measurements confirm the formation of charge carriers that can participate in the redox transformations of organic compounds. It is important to note that in addition to an oxidative ability, the synthesized TiO_2_-N photocatalyst, supported with noble metals, can provide the photocatalytic evolution of molecular hydrogen from water. Details and kinetic curves of hydrogen evolution for TiO_2_-N supported with several metals are shown in Appendix A.

Figure 6b shows the action spectra of TiO_2_ and TiO_2_-N samples as a function of photonic efficiency in the reaction of acetone oxidation on the basic wavelength of the LED source used for irradiation of photocatalysts. The spectra obtained correlate well to the absorption spectra of the studied samples (Figure 2a). The TiO_2_ sample synthesized without ammonia exhibits photocatalytic activity under UV light, while it only slightly utilizes visible light in the region up to 424 nm with extremely low efficiency. The TiO_2_-N sample exhibits a similar trend in the UV region, but the corresponding values of photonic efficiency are substantially higher than those for the TiO_2_ sample. For example, the photonic efficiencies of acetone oxidation under an LED with a basic wavelength of 367 nm were 69 and 33% for TiO_2_-N and TiO_2_, respectively. The lower activity of the TiO_2_ sample is probably due to strongly adsorbed Na^+^ ions, which can remain on the TiO_2_ surface after synthesis with NaOH. Sodium ions are well known to play the role of sites for the recombination of photogenerated charge carriers. In contrast to TiO_2_, N-doped TiO_2_ exhibits activity under visible light with wavelengths up to 525 nm. A local maximum of the activity of TiO_2_-N is observed at ca. 450 nm. It corresponds to the maximum in its absorption spectrum (Figure 2a). The value of photonic efficiency corresponding to this wavelength is found to be 3.8%.

Therefore, the formation of reactive oxygen species occurs on the irradiated surface of TiO_2_-N, and the formed species can oxidize adsorbed organic contaminants. Both types of data in Figure 6 show that the efficiency of light utilization by TiO_2_-N in the visible region is lower than that in the UV region, which is attributed to the region of fundamental absorption in TiO_2_. As mentioned in the Introduction, the content of UV light in the total solar radiation is low due to many photons corresponding to the visible region. The electricity-to-light efficiency of visible LEDs (e.g., blue LEDs) is substantially higher than the efficiency of UV LEDs. As a result, blue LEDs provide a substantially higher photon flux than UV LEDs at the same electric power. Thus, in the cases of both natural and artificial light sources, a higher number of photons in the visible region compared to the UV region would provide an increased rate of photocatalytic reaction that can minimize the difference in the photocatalytic ability of TiO_2_-N under UV and visible light despite different values of efficiency. This indicates the potential of TiO_2_-N for application in visible-light-driven processes.

Before considering the effect of preparation conditions, it should be noted that one of the key steps during the synthesis of TiO_2_-N from the TiOSO_4_ precursor is washing titanium oxide hydrate after its precipitation due to a high amount of sulfur species in the precursor. For instance, Figure 7a shows the DRIFT spectra of two synthesized TiO_2_-N samples, which differ in washing procedure, and Figure 7b shows the steady-state photocatalytic activity of these samples in the oxidation of acetone vapor under blue light. Three intense absorption bands in the region of 1000–1300 cm^−1^ are clearly observed in the DRIFT spectrum of the sample prepared without washing (Figure 7a). These bands can be attributed to sulfates bonded to the surface of TiO_2_ [73]. This indicates that a high number of sulfates remain on the surface of photocatalysts prepared without or with poor washing because of a high affinity of the SO_4_^2−^-group for Ti^4+^ sites [55]. An extremely low activity under visible light was observed for the sample prepared without intermediate washing of the precipitate (Figure 7b).

Otherwise, multiple washes of the precipitate before its calcination at a high temperature resulted in the removal of sulfates from the surface of the sample because the mentioned bands in the DRIFT spectrum vanished (Figure 7a). As shown in Figure 7b, this procedure led to a 3-fold increase in the visible-light activity of this sample. Thus, removing sulfur species after hydrolysis of the TiOSO_4_ precursor is important for the preparation of highly active photocatalysts using the proposed technique. At the same time, even multiple washed photocatalysts can have a high amount of residual sulfur. According to CHNS analysis, the S content in the final sample can be up to 4 wt.% depending on other preparation conditions, as will be discussed below. This is a reason that the photocatalytic activity of materials prepared from titanium oxysulfate of various suppliers can differ substantially from each other.

The temperature used for the calcination of precipitate formed after hydrolysis of TiOSO_4_ strongly affects the functional properties of synthesized TiO_2_-N. As shown in Figure 8a, the photocatalytic activity of TiO_2_-N samples under UV light increases monotonically as the calcination temperature increases. This behavior can be attributed to an enhancement in the crystallinity of samples that has a strong positive effect. An intense crystallization of TiO_2_ during the increase in the calcination temperature of samples from 250–300 °C to 350 °C would be a reason for the substantial increase in their activities. As mentioned above, a further increase in the calcination temperature increased the extent of crystallinity, which means a lower number of defects in the crystal lattice of TiO_2_. Defects in TiO_2_ are sites where the recombination of electrons and holes occurs. Therefore, a lower number of defects provides a lower rate of recombination and, consequently, a higher rate of photocatalytic reaction [74]. Comparing the best activity of TiO_2_-N samples under UV light (2.8 µmol min^−1^) with corresponding data for commercial photocatalysts TiO_2_ P25 (1.3 µmol min^−1^) and KC (2.3 µmol min^−1^) under similar conditions, TiO_2_-N provides the highest oxidation rate in the studied test reaction. This suggests that the proposed preparation technique facilitates the production of highly active photocatalysts for processes driven under UV light (e.g., for conventional photocatalytic air purifiers equipped with UV light sources).

A different pattern was observed in the case of acetone oxidation under blue light. Figure 8a shows that the visible-light activity of TiO_2_-N samples has a dome-shaped dependence on the calcination temperature. Similar to UV light, a substantial increase in the activity of samples was observed at a change in calcination temperature from 250–300 °C to 350 °C. This fact correlates with the crystal data of the samples (Figure 1a) and can be attributed to an intense crystallization of TiO_2_ in the mentioned range of temperatures. The visible-light activity slightly decreased in the temperature range of 350–500 °C, but it finally dropped down at the calcination temperature of 600 °C.

The visible-light activity of TiO_2_ photocatalysts prepared under reducing conditions is commonly attributed to the reduced states of titanium in the TiO_2_ lattice (e.g., Ti^3+^) because the energy levels corresponding to these states are located lower than the CB of TiO_2_ [75,76,77]. According to the results of XPS and EPR analyses (Figure 4 and Figure 5), the prepared TiO_2_-N samples contain similar centers. We checked the effect of calcination temperature on the concentration of these centers using the EPR technique. Figure 8b shows a change in the intensity of the EPR signal with a g-factor of 1.99, attributed to Ti^3+^ centers, when the calcination temperature is changed from 250 °C to 600 °C. This plot clearly indicates that the photocatalytic activity of TiO_2_-N under visible light does not correlate with the concentration of localized electron centers Ti^3+^ because the intensity of the mentioned signal after 350 °C has a similar level and does not decrease even at 600 °C. This confirms that the visible-light activity of TiO_2_-N cannot be attributed completely to the Ti^3+^ states.

However, the visible-light activity does not also correlate with the total content of nitrogen in the prepared samples. As shown in Table 1, the nitrogen content decreases monotonically from 0.8 wt.% to the limit of detection as the calcination temperature increases from 250 °C to 600 °C. This means that not all nitrogen species in TiO_2_-N respond to visible-light activity. According to the analysis of DRIFT spectra, the uncalcined TiO_2_-N sample formed after mixing the reagents contains a large amount of ammonia adsorbed on the surface of TiO_2_, which results in a high value of nitrogen content as estimated using CHNS analysis. Calcination of the TiO_2_-N sample leads to the oxidation of ammonia. Various oxidized forms of nitrogen were detected on the surface of samples calcined at a low temperature. A discussion of nitrogen species on the surface of the photocatalyst based on the results of DRIFT analysis is provided in the Appendix A. Therefore, it can be assumed that calcination leads to the transformation of some nitrogen species into a state that can provide a visible-light response and photocatalytic activity under visible light. The actual value of temperature affects the amount of target nitrogen species. This result agrees with the temperature dependence of light absorption shown in Figure 3b.

The EPR technique allows us to identify the paramagnetic nitrogen species. Figure 8b shows that the intensity of the EPR signal, attributed to the N-based light-induced paramagnetic species (see Table 2), also has a dome-shaped dependence on the calcination temperature with a maximum at 350–400 °C, which is similar to the plots of light absorption and visible-light activity mentioned above. A further increase in the calcination temperature leads to a monotonic decrease in the intensity of the EPR signal. For the sample calcined at 600 °C, the intensity decreases by almost two orders of magnitude compared to the maximum value. These data confirm that the nitrogen paramagnetic species respond to the activity of TiO-N photocatalysts under visible light.

Therefore, based on the results of characterization techniques, the existence of an optimum calcination temperature can be attributed to the incorporation of nitrogen in the TiO_2_ lattice at an increased temperature. This results in the creation of impurity energy levels in the TiO_2_ band gap and provides the absorption of light in the visible region of the spectrum. When the calcination temperature is too high, nitrogen is oxidized and removed from the structure. This leads to a decrease in the photocatalytic activity under visible light. It can be assumed that Ti^3+^ centers contribute to the photocatalytic activity under visible light, especially for the samples calcined at high temperatures.

Considering the highest activity of TiO_2_-N under blue light (1.2 µmol min^−1^) with the corresponding data for the commercial photocatalysts TiO_2_ P25 (0.03 µmol min^−1^) and TiO_2_ KC (0.07 µmol min^−1^) under similar conditions, TiO_2_-N provides the highest oxidation rate in the studied test reaction. This proves that the proposed preparation technique allows the production of highly active N-doped TiO_2_, which is able to perform photocatalytic processes under both UV and visible light. As additional benchmarks, Appendix A shows a comparison of the visible-light photocatalytic activity of synthesized TiO_2_-N with other TiO_2_-N photocatalysts described in previously published papers.

### 3.2. Enhancement of Stability Via Surface Modification

The stability of photocatalysts under long-term irradiation is an important requirement for practical applications. One of the drawbacks of pristine TiO_2_-N is that its initial activity decreases gradually under long-term exposure to high power radiation. Under the conditions of this study (blue light, 164 mW cm^−2^), the rate of acetone oxidation over TiO_2_-N decreased by 40% after irradiation for 6 h and by 5% more after irradiation for an additional 24 h (Figure 9).

According to a previously published study [78], this deactivation occurs due to the degradation of nitrogen species by photogenerated holes, which do not migrate to the surface of the photocatalyst. In our case, this was confirmed by a decrease in the intensity of nitrogen signals in XPS and EPR spectra for the sample studied after the stability test. The intensity of the EPR signal attributed to nitrogen decreased by 66% after the experiment on stability, which indicates a partial degradation of visible-light responsive nitrogen species by the photogenerated holes. However, after the period mentioned above (i.e., 30 h), the visible-light activity of TiO_2_-N reached a permanent level and did not change substantially under further irradiation.

Recently, we have shown [79] that the combination of TiO_2_-N with a Bi_2_WO_6_ semiconductor can provide a nanocomposite photocatalyst, which maintains a high level of activity even under high power radiation. Herein, we show another example of the modification of TiO_2_-N to enhance its stability under long-term exposure to power radiation. It is a simple modification of TiO_2_-N with copper species.

Copper species were deposited on the surface of TiO_2_-N via the impregnation method. A Cu loading of 1 wt.% was selected as an optimum value based on the results of a preliminary study concerning the effect of Cu content on the activity [80]. Cu-modified TiO_2_-N was studied using XRD, XPS and EPR techniques. There was no substantial difference between the XRD patterns of the pristine and Cu-modified TiO_2_-N samples probably due to the low Cu content. Figure 10a,b shows the Cu 2p and Cu LMM spectral regions of the synthesized Cu/TiO_2_-N sample. The absence of noticeable shake-up satellites in the region of 938–945 eV in the Cu 2p spectrum indicates a reduced state of copper, excluding Cu^2+^ [81]. The position of the Cu 2p_3/2_ peak near 932.5 eV indicates the possibility for the presence of copper in the form of Cu^0^ or Cu^1+^ [82]. For accurate identification, the modified Auger parameter of copper (α’-Cu) was determined as the sum of the binding energy of the Cu 2p_3/2_ peak maximum and the kinetic energy of the Cu L_3_M_45_M_45_ maximum. The estimated value of the α’-Cu parameter was 1847.8 eV, which corresponds well to the Cu^1+^ state [83]. However, the EPR spectra of Cu/TiO_2_-N in Figure 10c,d show a prominent signal at g = 2.1 corresponding to Cu^2+^ ions in octahedral coordination [84,85].

A contradiction in the XPS and EPR data may be due to the reduction of copper species under X-ray radiation and the high stability of the Cu^1+^ state on the surface of TiO_2_ [86]. The pretreatment of the sample with X-ray radiation for 1 h was necessary to obtain high-quality XPS spectra. This pretreatment could reduce all copper species on the TiO_2_-N surface to the Cu^1+^ state, while the vacuum conditions did not allow Cu to be oxidized back to the Cu^2+^ state. On the other hand, Cu^1+^ species cannot be detected using the EPR technique because they are diamagnetic. As a result, only the Cu^2+^ state is detected in the EPR spectra. According to a previously published study [87], many different states of copper (Cu^0^, Cu_x_O, Cu^2+^) can be found on the surface of TiO_2_, and the ratio between these states can be substantially changed under radiation. This confirms the possibility of various copper species on the surface of TiO_2_-N.

Iron species were used as a counterexample to copper species because it is well known that the energy levels of Fe^3+^/Fe^2+^ species grafted at the TiO_2_ surface are located slightly lower than the CB minimum of TiO_2_ and, consequently, can participate in the transfer of photogenerated electrons only [88,89]. Fe content of 0.1 wt.% was selected as an optimum value based on the results of a preliminary study concerning the effect of Fe content on the activity (see Appendix A). Reversible transfer between the charge states of Fe^3+^ and Fe^2+^ commonly occurs in photocatalytic reactions catalyzed by Fe/TiO_2_ in the presence of oxygen. In our case, XPS analysis of the synthesized Fe/TiO_2_-N sample shows only the Fe^2+^ state (see Appendix A). This situation is similar to the case of the copper species discussed above. A partial reduction of iron species until the Fe^2+^ state occurs under X-ray irradiation under vacuum conditions during the analysis. The absence of oxygen prevents the reoxidation of Fe^2+^ to Fe^3+^.

Figure 9a shows the effect of TiO_2_-N modification with iron and copper species on the visible-light activity and stability of the photocatalyst. The surface modification of TiO_2_-N with iron increased its initial activity by 30%. This effect is commonly attributed to an enhanced separation of charge carriers due to the localization of electrons in Fe species on the surface of the photocatalyst. However, the time dependence of the activity for Fe-modified TiO_2_-N is similar to that for pristine TiO_2_-N. The activity of Fe/TiO_2_-N decreased by 39% after irradiation for 19 h, which is close to the data for TiO_2_-N (Figure 9b). This behavior can be explained by the fact that iron species can capture electrons from TiO_2_-N, but they have a low effect on the transfer of photogenerated holes [90]. Therefore, no enhancement in the stability of TiO_2_-N is observed upon its modification with iron species.

A similar increase in the initial activity of TiO_2_-N (~30%) was also observed upon modification with copper species. However, in contrast to iron, the visible light activity of Cu-modified TiO_2_-N did not change substantially during long-term irradiation. The total decrease in the activity after 30 h of irradiation was lower than 10% (Figure 9b). According to the XPS data, no changes in the states of nitrogen and copper were found. According to the EPR data, no changes are also observed for the state of nitrogen paramagnetic species, but the intensity of the corresponding signal decreased compared to the initial value. However, the decrease in the signal intensity was only 33%, which is much lower than the corresponding value (66%) for pristine TiO_2_-N. The data discussed above allow us to assume that copper species affect the pathway of holes. As a confirmation of this statement, we have previously studied the action spectrum of Cu-grafted TiO_2_ and showed that this photocatalyst exhibits activity in the region up to 500 nm only, which may indicate interfacial charge transfer from TiO_2_ to Cu_x_O clusters as the major pathway of photoexcitation in the concerned system under visible light [80]. This means that the energy levels of copper species are located above the VB of TiO_2_, and they can participate in the interfacial charge transfer from the nitrogen levels in TiO_2_.

Based on the experimental results and literature data, it can be assumed that copper species enhance the transfer of photogenerated holes and their localization on the surface of TiO_2_-N, as well as further interfacial transfer to electron donors. If the transfer of holes is considered as the bottleneck of oxidation processes [6], an enhanced hole transfer would improve the rate of the whole process and, consequently, decrease the probability of degradation of the nitrogen species in TiO_2_.

Figure 11 shows the proposed energy diagram, which illustrates the pathways of photogenerated charge carriers for all considered systems. Irradiation of TiO_2_-N results in excitation of electrons to the CB. These photogenerated electrons can further interact with an acceptor (A), resulting in its reduction (A^−^). In the case of the Fe/TiO_2_-N system, the electrons can first transfer to the energy levels of Fe species and further interact with the acceptor. The photogenerated holes in both TiO_2_-N and Fe/TiO_2_-N systems can interact with a donor (D), resulting in its oxidation (D^+^). Localization of photogenerated holes in the energy levels of nitrogen species (NO) leads to their partial degradation and conversion to a non-paramagnetic form (NO^+^). However, in the case of the Cu/TiO_2_-N system, the photogenerated holes can first transfer to the energy levels of Cu species, thus reducing the degradation of nitrogen species (NO), and further interact with the donor (D), resulting in its oxidation (D^+^).

Highly active N-doped TiO_2_ photocatalysts with a visible-light response have potential in the design of nanocomposite materials with other semiconductors, which have a narrower band gap, for better utilization of visible light due to excitation of electrons in both components and realization of interfacial charge transfer according to the heterojunctions of favorable types. In this paper, we proposed a preparation technique that can be applied for the large-scale production of highly active and stable N-doped TiO_2_ for use in visible-light-driven processes.

## 4. Conclusions

Titanium oxysulfate (TiOSO_4_) can be successfully used as a low-cost inorganic precursor for the preparation of highly active photocatalysts sensitive to visible light. A pH-controlled precipitation of TiO_2_ from an aqueous solution of TiOSO_4_ using ammonia followed by the calcination of precipitate in air at an increased temperature results in the formation of nanocrystalline anatase, which can absorb radiation in the region up to 540 nm due to the presence of nitrogen impurities in TiO_2_. The as-prepared N-doped TiO_2_ photocatalyst can completely oxidize volatile organic compounds under UV and visible light in the corresponding region. The UV-light activity of N-doped TiO_2_ monotonically increases as the calcination temperature increases, which is attributed to an improvement in the crystallinity of TiO_2_. Otherwise, the visible-light activity has a dome-shaped dependence on the temperature with a maximum at 350–400 °C due to the removal of nitrogen species from TiO_2_ at higher temperatures. The absorption of visible light and visible-light activity of the prepared N-doped TiO_2_ correlate with the concentration of paramagnetic nitrogen species detected using the EPR technique. Residual sulfates on the surface of the photocatalyst can substantially decrease its visible-light activity; therefore, washing the precipitate is a key step during the synthesis of N-doped TiO_2_ from the TiOSO_4_ precursor. Copper species on the surface of N-doped TiO_2_ oppositely increase its activity under visible light and prevent intense deactivation under radiation of a high power that can be attributed to an enhanced transfer of charge carriers.

## Figures and Tables

**Figure 1 nanomaterials-12-04146-f001:**
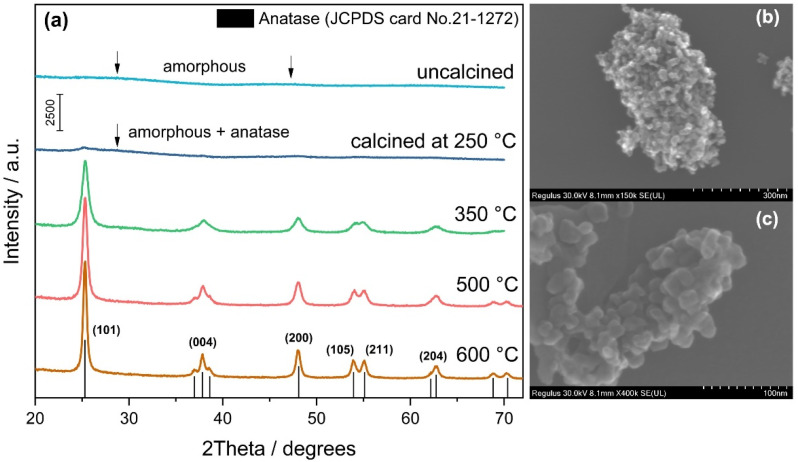
XRD patterns of TiO_2_-N samples calcined at different temperatures (**a**) and SEM micrographs of the sample calcined at 350 °C (**b**,**c**).

**Figure 2 nanomaterials-12-04146-f002:**
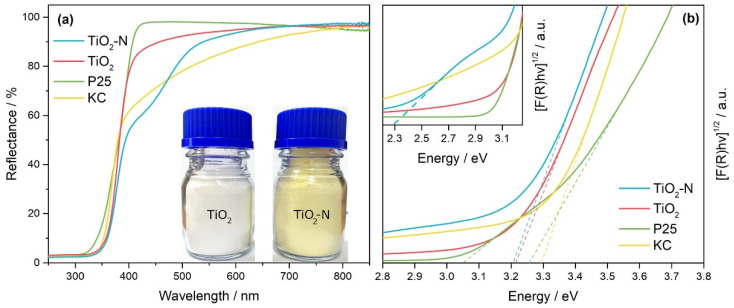
UV–vis DRS spectra (**a**) and Tauc plots (**b**) for TiO_2_-N, TiO_2_, P25, and KC photocatalysts. The inset in (**b**) shows an enlarged area of Tauc plots in the range of 2.2–3.25 eV.

**Figure 3 nanomaterials-12-04146-f003:**
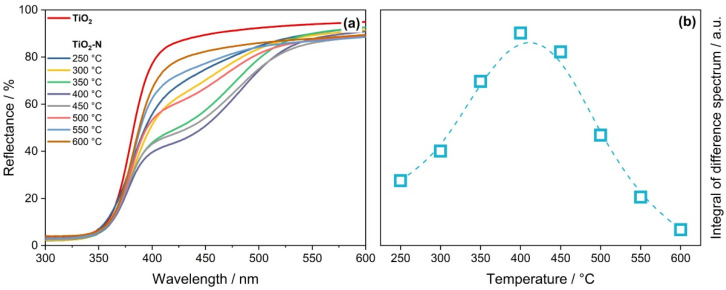
Effect of the calcination temperature on UV–vis DRS spectra of TiO_2_-N (**a**) and integral of the difference spectrum between TiO_2_-N and TiO_2_ in the region of 380–550 nm (**b**).

**Figure 4 nanomaterials-12-04146-f004:**
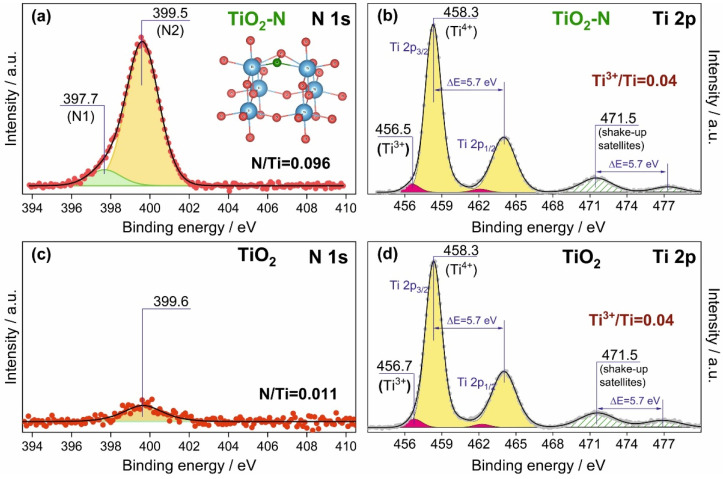
Photoelectron N 1s (**a**,**c**) and Ti 2p (**b**,**d**) spectral regions for TiO_2_-N and TiO_2_ samples synthesized at 350 °C.

**Figure 5 nanomaterials-12-04146-f005:**
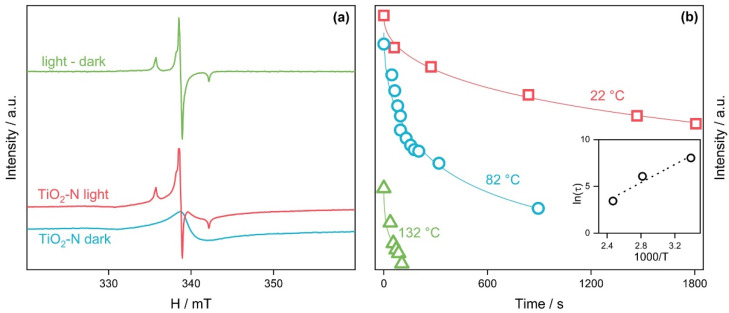
EPR spectra of TiO_2_-N in the dark or under light exposure (**a**) and decay plots for the intensity of the EPR signal at different temperatures (**b**). The inset in (**b**) shows the temperature dependence of the average decay time.

**Figure 6 nanomaterials-12-04146-f006:**
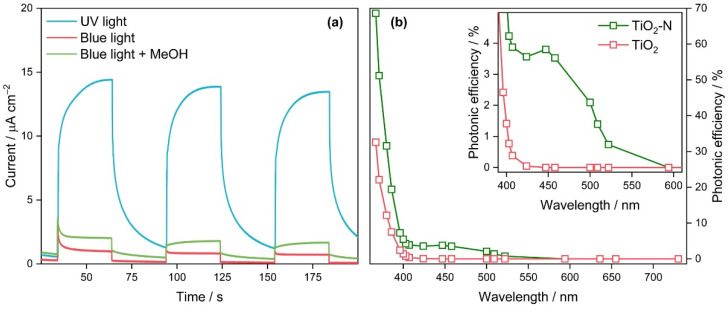
Photocurrent of TiO_2_-N under UV and visible light (**a**); action spectra of TiO_2_-N and TiO_2_ synthesized at 350 °C (**b**). The inset in (**b**) shows an enlarged area of action spectra in the range of 380–620 nm.

**Figure 7 nanomaterials-12-04146-f007:**
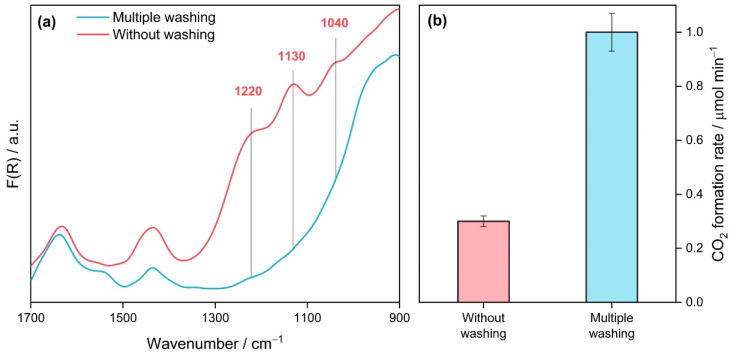
DRIFT spectra (**a**) and photocatalytic activity (**b**) of TiO_2_-N samples (350 °C) with and without intermediate washing of the precipitate.

**Figure 8 nanomaterials-12-04146-f008:**
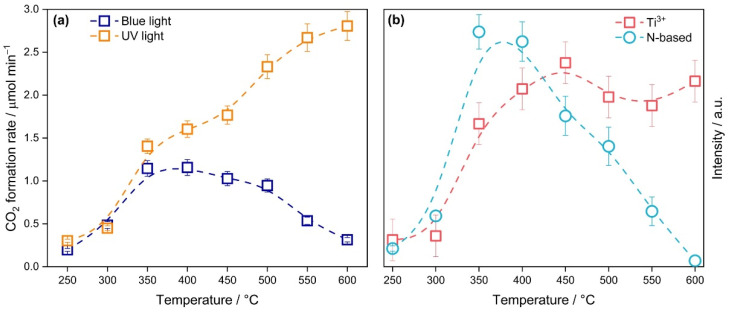
Effect of calcination temperature on the photocatalytic activity (**a**) and intensity of EPR signals (**b**) of the TiO_2_-N sample.

**Figure 9 nanomaterials-12-04146-f009:**
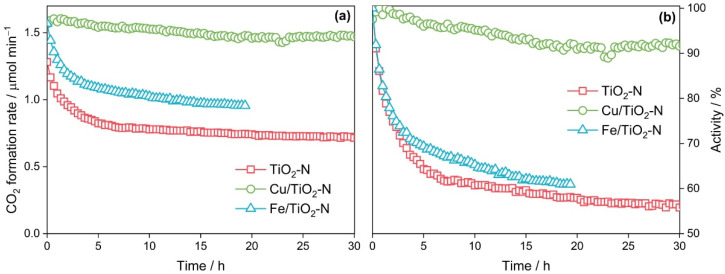
Effect of copper and iron species on the stability of TiO_2_-N (350 °C) in absolute (**a**) and relative (**b**) coordinates.

**Figure 10 nanomaterials-12-04146-f010:**
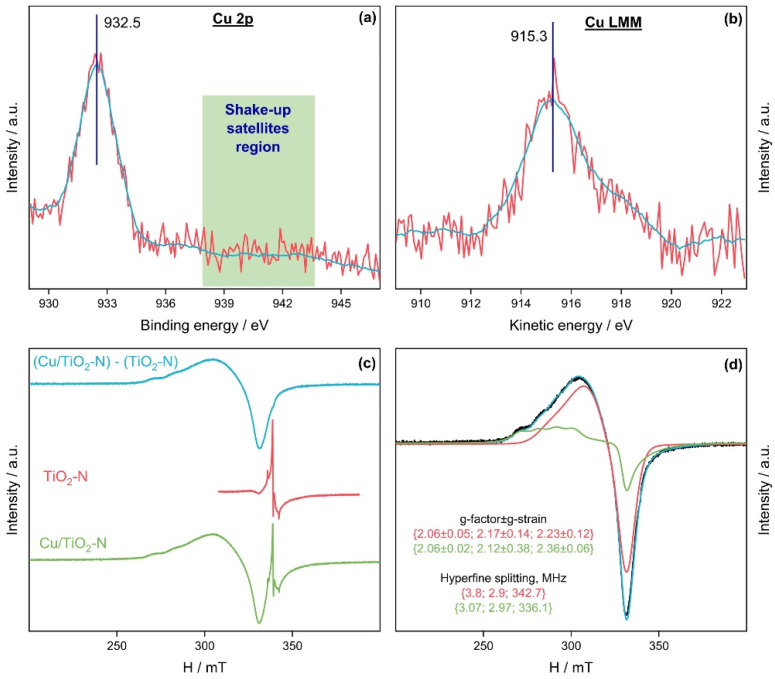
Photoelectron Cu 2p (**a**) and Cu LMM (**b**) spectral regions for the synthesized Cu/TiO_2_-N sample; experimental (**c**) and simulated (**d**) EPR spectra of Cu/TiO_2_-N.

**Figure 11 nanomaterials-12-04146-f011:**
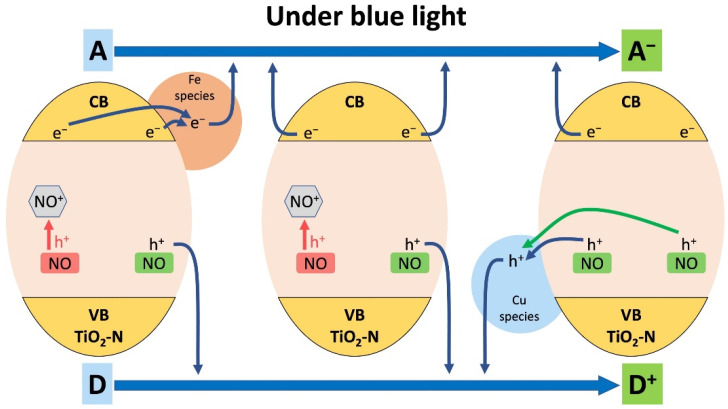
Energy diagram for Fe/TiO_2_-N (**left**), TiO_2_-N (**center**), and Cu/TiO_2_-N (**right**).

**Table 1 nanomaterials-12-04146-t001:** Characteristics of the samples calcined at different temperatures.

Calcination Temperature, °C	Crystal Size, nm	Stain Values (ε)	Surface Area, m^2^ g^−1^	Pore Volume, cm^3^ g^−1^	Pore Diameter, Å	Nitrogen Content, wt.%	Sulfur Content, wt.%	Band Gap, eV	MEE **, eV
250	n.d. *	n.d.	344	0.59	69	0.8 ± 0.1	0.14 ± 0.04	3.22	2.22
300						0.6 ± 0.2	0.16 ± 0.06	3.20	2.27
350	16	0.0031	151	0.43	115	0.5 ± 0.1	0.16 ± 0.04	3.19	2.32
400						0.4 ± 0.1	0.16 ± 0.06	3.17	2.30
450	20	0.0020	113	0.38	136	0.36 ± 0.03	0.11 ± 0.03	3.18	2.27
500	23	0.0017				0.27 ± 0.03	0.08 ± 0.01	3.19	2.24
550			73	0.31	172	0.14 ± 0.01	0.14 ± 0.04	3.19	n.d.
600	25	0.0010				n.d.	0.11 ± 0.03	3.18	n.d.

* n.d. = non-detectable. ** MEE = minimum energy required for the photoexcitation of electron.

**Table 2 nanomaterials-12-04146-t002:** Parameters of individual components in EPR spectra.

Paramagnetic Center	g-Factor ± g-Strain	Hyperfine Splitting, MHz
Ti^3+^	1.99	
N	{2.003 ± 0.001; 2.004 ± 0.008; 2.002 ± 0.006}	{2.3; 12.3; 88.9}

## Data Availability

The data presented in this study are available on request from the corresponding author. The data are not publicly available due to privacy.

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
