# Peer review of "Visible-Light-Active N-Doped TiO_2_ Photocatalysts: Synthesis from TiOSO_4_, Characterization, and Enhancement of Stability Via Surface Modification"

_nanomaterials, 2022, doi:10.3390/nano12234146_

Round 1

Reviewer 1 Report

Comments

In this research, the titanium oxy-sulfate (TiOSO4) was used as a precursor and ammonia as a precipitant, forming N-doped TiO2 and forming nanocrystalline anatase TiO2 by regulating the PH value, and enabling complete oxidation of volatile organic compounds under UV and visible light. However, the following set of issues need to be addressed before public publication.

1.      The keyword "long-term stability" was not mentioned in the abstract, suggesting that the authors reconsider the choice of keywords.

2.      There should be spaces between “%/” and the number, and the author needs to carefully check the format in the text as well as other grammar problems.

3.      There are details problem in the text and supporting information (such as: Figure 3, Figure S3, etc.). In addition, in the description section of the supporting Information there exist many " ". This is unreasonable, and the author needs to carefully check and correct it.

4.      The crystal surface corresponding to the peak should be marked in the XRD diagram, and the abscissa of different samples should be consistent, furthermore, the line color should be consistent with the Figure 3a.

5.      The XPS of TiO2 should be given in the XPS analysis and compared with TiO2-N.

6.      For the Figure 11 energy diagram, the authors should give a more detailed description of the transfer path of all studied systems.

7.      Some recent works focus on the studying of metal oxide-based photocatalyst are suggested to cited and review for polishing the revised manuscript, such as Journal of Colloid and Interface Science 602 (2021) 889–897

Author Response

 We would like to thank the reviewers for constructive remarks, which help us to improve the manuscript. Please find the detailed answers and comments below.

Comment: In this research, the titanium oxy-sulfate (TiOSO4) was used as a precursor and ammonia as a precipitant, forming N-doped TiO2 and forming nanocrystalline anatase TiO2 by regulating the PH value, and enabling complete oxidation of volatile organic compounds under UV and visible light. However, the following set of issues need to be addressed before public publication.

Response: Thank you for your appreciation of our work and your valuable comments that helped us improve this paper. A detailed response to each comment is provided below.

Comment: 1. The keyword "long-term stability" was not mentioned in the abstract, suggesting that the authors reconsider the choice of keywords.

Response: According to your recommendation, the keyword “long-term stability” was changed to “stability” because the abstract contains a mention of this aspect for the synthesized photocatalysts.

Comment: 2. There should be spaces between “%/℃” and the number, and the author needs to carefully check the format in the text as well as other grammar problems.

Response: We completely agree that a space should be between the number and “℃” symbol, and this style of representation is used over the whole text of manuscript.

In the case of “%” symbol (similarly to “°”, “"”, etc), no spaces are typically used (e.g., Nature Catal. 2 (2019) 387–399, J. Col. Inter. Sci. 602 (2021) 889–897). We also preferer this type of representation, therefore there is no space between “%” symbol and number in the text of manuscript.

The manuscript was thoroughly checked again to avoid other problems. Please find the revised version of manuscript.

Comment: 3. There are details problem in the text and supporting information (such as: Figure 3, Figure S3, etc.). In addition, in the description section of the supporting Information there exist many " ". This is unreasonable, and the author needs to carefully check and correct it.

Response: We have added to Experimental section a detailed information on the calculation of difference UV-vis spectra and their integration to analyze the effect of calcination temperature on absorption of light. We have added a detailed information to the text in Supplementary. Please find the revised versions of these documents.

Comment: 4. The crystal surface corresponding to the peak should be marked in the XRD diagram, and the abscissa of different samples should be consistent, furthermore, the line color should be consistent with the Figure 3a.

Response: We have modified the Figure 1 according to your recommendation. Please find the revised version of manuscript.

Comment: 5. The XPS of TiO2 should be given in the XPS analysis and compared with TiO2-N.

Response: We have added XPS spectra of TiO2 to the Figure 4 and compared them with corresponded spectra of TiO2-N in the text of manuscript. Please find the revised version.

Comment: 6. For the Figure 11 energy diagram, the authors should give a more detailed description of the transfer path of all studied systems.

Response: According to your recommendation, we more discussed the proposed energy diagram. Please find the revised version of manuscript.

Comment: 7. Some recent works focus on the studying of metal oxide-based photocatalyst are suggested to cited and review for polishing the revised manuscript, such as Journal of Colloid and Interface Science 602 (2021) 889–897

Response: We have referred to the proposed article in the Introduction of revised paper.

Comment: Extensive editing of English language and style required.

Response: English language in the manuscript was checked again and edited using AJE digital editing service (https://www.aje.com/). The revised version of manuscript was also checked using AJE Grammar Check tool (https://www.aje.com/grammar-check/). This tool shows 8.0/10 score (91st percentile of papers submitted to AJE) and indicates that the manuscript is well written and does not need extensive language editing. Final polishing will be made at proof-reading step.

Reviewer 2 Report

Authors reported a low-cost method to prepare N-doped TiO2, which could provide the complete oxidation of volatile organic compounds both under UV and visible light.  Moreover, copper species were used to decorate N-doped TiO2 to improve its activity and stability. This work made a full study on visible-light-active N-doped TiO2 photocatalysts after the following concerns are addressed. 

1. XPS showed that nitrogen were mainly incorporated into interstitial positions of TiO2. If so, the XRD peak of N-doped TiO2 might be shifted compared to TiO2. Authors should further verify it. 

2. Why copper species can enhance the transfer of photogenerated holes, while iron species can only enhance the transfer of photogenerated electrons? Authors should provide more evidences.

3. Author can consider the following related references about N-doped TiOor Ti3+ defects1) Jian Yuan, etc. Preparations and photocatalytic hydrogen evolution of N-doped TiO2 from urea and titanium tetrachloride, International Journal of Hydrogen Energy, 31(10), 2006. 2) Yang Yang, etc. Enhanced photocatalytic overall water splitting by tuning the relative concentration ratio of bulk defects to surface defects in SrTiO3, International Journal of Hydrogen Energy, 2022.

Author Response

 We would like to thank the reviewers for constructive remarks, which help us to improve the manuscript. Please find the detailed answers and comments below.

Comment: Authors reported a low-cost method to prepare N-doped TiO2, which could provide the complete oxidation of volatile organic compounds both under UV and visible light.  Moreover, copper species were used to decorate N-doped TiO2 to improve its activity and stability. This work made a full study on visible-light-active N-doped TiO2 photocatalysts after the following concerns are addressed.

Response: Thank you for your appreciation of our work and your valuable comments that helped us improve this paper. A detailed response to each comment is provided below.

Comment: 1. XPS showed that nitrogen were mainly incorporated into interstitial positions of TiO2. If so, the XRD peak of N-doped TiO2 might be shifted compared to TiO2. Authors should further verify it.

Response: Yes, the diffraction peaks of N-doped TiO2 is slightly shifted compared to TiO2 sample because the constantsof tetragonal lattice (i.e., a and c) and, consequently, the volume of the unit cell (V) are changed. There is a trend that a greater content of nitrogen leads to a higher value of strains (i.e., microdeformations) but a lower value of unit cell volume (V). We have added this information to the main text of manuscript and to the Table S1 in Supplementary. Please find the revised version of manuscript.

Comment: 2. Why copper species can enhance the transfer of photogenerated holes, while iron species can only enhance the transfer of photogenerated electrons? Authors should provide more evidences.

Response: In this study, iron species were used as a counterexample to copper species because it is well known in literature that the energy levels of Fe species are located near the CB minimum of TiO2 and, as a result, can participate in the transfer of photogenerated electrons only. In contrast, the energy levels of copper species are located much lower, and some of them can give electrons to the photogenerated holes localized at nitrogen levels (or, in other words, the holes are transferred from the nitrogen levels to the energy levels of copper species).

We referred to the corresponding papers in the text of manuscript for supporting and illustrating the mentioned pathways. Please find the revised version of manuscript.

Comment: 3. Author can consider the following related references about N-doped TiOor Ti3+ defects: 1) Jian Yuan, etc. Preparations and photocatalytic hydrogen evolution of N-doped TiO2 from urea and titanium tetrachloride, International Journal of Hydrogen Energy, 31(10), 2006. 2) Yang Yang, etc. Enhanced photocatalytic overall water splitting by tuning the relative concentration ratio of bulk defects to surface defects in SrTiO3, International Journal of Hydrogen Energy, 2022.

Response: We have referred to the proposed articles in Introduction. Please find the revised version of manuscript.